# *Poria cocos* Polysaccharide Ameliorated Antibiotic-Associated Diarrhea in Mice via Regulating the Homeostasis of the Gut Microbiota and Intestinal Mucosal Barrier

**DOI:** 10.3390/ijms24021423

**Published:** 2023-01-11

**Authors:** Huachong Xu, Shiqi Wang, Yawen Jiang, Jialin Wu, Lili Chen, Yujia Ding, Yingtong Zhou, Li Deng, Xiaoyin Chen

**Affiliations:** 1College of Traditional Chinese Medicine, Jinan University, Guangzhou 510632, China; 2Guangzhou Key Laboratory of Formula-Pattern of Traditional Chinese Medicine, Guangzhou 510632, China

**Keywords:** *Poria cocos* polysaccharide, antibiotic-associated diarrhea, gut microbiota, 16S rRNA sequencing, antibiotics

## Abstract

*Poria cocos* polysaccharides (PCP) have been validated for several biological activities, including antitumor, anti-inflammatory, antioxidant, immunomodulatory, hepatoprotective and modulation on gut microbiota. In this research, we aim to demonstrate the potential prebiotic effects and the therapeutic efficacies of PCP in the treatment of antibiotic-associated diarrhea (AAD), and confirm the beneficial effects of PCP on gut dysbiosis. Antibiotic-associated diarrhea mice models were established by treating them with broad-spectrum antibiotics in drinking water for seven days. Mice in two groups treated with probiotics and polysaccharide were given Bifico capsules (4.2 g/kg/d) and PCP (250 mg/kg/d) for seven days using intragastric gavage, respectively. To observe the regulatory effects of PCP on gut microbiota and intestinal mucosal barrier, we conducted the following experiments: intestinal flora analysis (16S rDNA sequencing), histology (H&E staining) and tight junction proteins (immunofluorescence staining). The levels of mRNA expression of receptors associated with inflammation and gut metabolism were assessed by real-time reverse transcription-polymerase chain reaction (RT-PCR). The study revealed that PCP can comprehensively improve the clinical symptoms of AAD mice, including fecal traits, mental state, hair quality, etc., similar to the effect of probiotics. Based on histology observation, PCP significantly improved the substantial structure of the intestine of AAD mice by increasing the expression levels of colonic tight junction protein zonula-occludens 1 (ZO-1) and its mRNA. Moreover, PCP not only increased the abundance of gut microbiota, but also increased the diversity of gut microbiota in AAD mice, including alpha diversity and beta diversity. Further analysis found that PCP can modulate seven characteristic species of intestinal flora in AAD mice, including *Parabacteroides_distasonis*, *Akkermansia_muciniphila*, *Clostridium_saccharolyticum*, *Ruminoc-occus_gnavus*, *Lactobacillus_salivarius*, *Salmonella_enterica* and *Mucispirillum_schaedleri.* Finally, enrichment analysis predicted that PCP may affect intestinal mucosal barrier function, host immune response and metabolic function by regulating the microbiota. RT-PCR experiments showed that PCP can participate in immunomodulatory and modulation on metabolic by regulating the mRNA expression of forkhead-box protein 3 (FOXP3) and G protein-coupled receptor 41 (GPR41). These results indicated that *Poria cocos* polysaccharide may ameliorate antibiotic-associated diarrhea in mice by regulating the homeostasis of the gut microbiota and intestinal mucosal barrier. In addition, polysaccharide-derived changes in intestinal microbiota were involved in the immunomodulatory activities and modulation of the metabolism.

## 1. Introduction

Intestinal microecology, as an extremely complex ecosystem, is mainly composed of intestinal microbes, intestinal mucosal barrier and intestinal mucosal immunity, and always maintains a relatively stable dynamic balance [1,2]. Normally, the gut microbiota can maintain the integrity of the host gut and regulate many important physiological functions, mainly including the homeostasis of energy metabolism and immune defense, thus affecting the body’s overall health [3,4]. Inevitably, however, intestinal microecology can be affected by a variety of external factors, especially direct exogenous factors such as climate environment, dietary habits, and medical medication, which can directly cause intestinal microbiota imbalance and lead to complex diseases and complications [5].

Currently, widely used antibiotics are a common factor causing intestinal dysbiosis, which directly destroys a large number of normal bacteria, thereby causing the dysregulation of intestinal bacteria and increasing the number of conditional pathogenic or drug-resistant bacteria [4]. Almost all antibiotics can cause clinical symptoms of diarrhea, known as antibiotic-associated diarrhea (AAD), by leading to dysfunction in the secretion, digestion and absorption of the gastrointestinal tract [6]. AAD can generally cause malnutrition, water and electrolyte disorders, acid-base imbalance and even secondary clostridium difficile infection, leading to pseudomembranous enteritis [7]. Intestinal microbiota imbalance is positively correlated with the duration of antibiotic use and antimicrobial spectrum, but negatively correlated with age [6]. AAD has complex and diverse mechanisms, but most researchers believe that it is mainly due to the destruction of intestinal microecology by antibiotics [8,9,10]. Hence, how to quickly restore the balance of intestinal microecology and prevent the destruction of antibiotics is extremely important for the clinical prevention and treatment of AAD. Normally, antibiotics, probiotics and fecal transplantation (FMT) are routine treatments for AAD, but they have limited efficacy and potential side effects [11]. Accordingly, the regulation effect of traditional Chinese medicine with the advantage of syndrome differentiation and treatment on intestinal flora provides a broader perspective for the treatment of AAD [12].

As the edible and medicinal mushroom, *Poria cocos*, known as Fu-ling, has long been applied in medical practice in traditional Chinese medicine and believed to have beneficial effects on food absorption and metabolism [13]. Therefore, *Poria cocos* is often used alone or in combination with other herbs (e.g., Si-Jun-Zi decoction and Shen-Ling-Bai-Zhu powder) to treat diarrhea. *Poria cocos* is commercially available and widely used in Asia for the formulation of nutraceuticals, dietary supplements, cosmetics and functional foods, especially in China [14]. Correspondingly, *Poria cocos* polysaccharides (PCP), which account for about 80% of dry sclerotia weight in *Poria cocos*, have been validated for several biological activities, including antitumor, anti-inflammatory, antioxidant, immunomodulatory and hepatoprotective [15]. Extensive studies have indicated that polysaccharides, as prebiotics, can promote the growth of beneficial bacteria and regulate the levels of metabolites of gut microbiota, for which the host physiology, particularly gastrointestinal health, can be improved. Previous studies have shown that several herbal polysaccharides can alleviate AAD based on modulating the gut microbiota [6,16,17,18,19], but the efficacy and mechanism of PCP in the treatment of AAD remain to be explored in more depth.

In this study, we demonstrated the potential prebiotic effects and the therapeutic efficacies of PCP in the treatment of AAD, and confirmed the beneficial effects of PCP on the antibiotic-associated gut dysbiosis and mucosal barrier damage.

## 2. Results

### 2.1. Animal Condition, Changes in Body Weight

In adaptive feeding deration, all mice survived well, with normal dietary intake, smooth and compliant hair, morphologically and moderately normal feces, good locomotor and good mental status. After the third day of modeling, mice treated with antibiotics developed diarrhea-related symptoms such as chills, poor diet, hair loss, dull coat color, yellow loose feces, anal prolapse, and listlessness, while the mice in normal group (NR) showed none of obvious abnormality, indicating that the AAD model was successfully established. After treatment, compared with the antibiotic-associated diarrhea group (AB), the diarrhea in the *Poria cocos* polysaccharides treatment group (PC) and probiotics treatment group (PB) mice was significantly reduced, with granular feces, thick-shiny hair, adequate intake, flexible movement and good mental state, suggesting polysaccharide and probiotics treatment can effectively improve the AAD mice diarrhea and related symptoms. Compared with the PB group, the improvement of diarrhea-related symptoms occurred earlier and was more significant in the PC group. All groups gained weight steadily from the start of the experiment, but from the 6th day of modeling, the mice in AB group consistently had the lowest average body weight (Figure 1A), indicating that diarrhea caused by antibiotics had a certain effect on body weight.

### 2.2. PCP Alleviates the Intestinal Mucosal Pathological Injury in AAD Mice

Antibiotics cause significant changes in the intestinal mucosal structure of the small intestine and colon, with enlargement of the cecum and an increase in intestinal fecal volume (Figure 1B). The structure of the small intestinal mucosa in the NR group was intact, with orderly and dense villi and deep crypts, and no ulcer or necrosis was observed. In the AB model group, the intestinal mucosa tissue was damaged, leading to the intestinal wall thinning, disordered and detached villi, and shallow crypts with inflammatory cell infiltration. Compared with the AB group, small intestinal mucosal structure in the PB group was relatively complete, mainly manifested as a slight disorder of the villi arrangement, almost no shedding, and partial alleviation of inflammatory infiltration (Figure 1C). The structure of small intestinal mucosa of the PB group was intact and basically returned to the same condition as that in the group NR (Figure 1C).

Similarly, antibiotics also increased the pathological damage of colonic tissue in mice. Compared to the NR group, the colon of mice in the AB group had far greater numbers of goblet cells, with increased mucus secretion, intestinal villi defects, central chylous duct marked dilatation and varying degrees of epithelial cell detachment (Figure 1D). The pathological changes of the colon in the PC and PB groups were significantly reduced compared with the model group (Figure 1D). These observations indicate that AAD mice can lead to the pathological damage of intestinal mucosa, and both PCP and probiotics can alleviate the pathological damage and relieve the symptoms related to diarrhea.

### 2.3. PCP Increases the Expression Intestinal Tight Junction Protein in AAD Mice

Tight junction proteins are an important component of intestinal mucosa, recognized as the basis for the normal physical and chemical barrier of intestinal mucosa. Here, immunofluorescence results showed that antibiotic modeling could induce a decrease in the expression of tight junction protein ZO-1 in the colon of AAD mice compared with the mice in the NR group (Figure 2A,B). Interestingly, the intervention of PCP and probiotics could reverse the tight junction protein loss caused by antibiotics, significantly increase the colonic ZO-1 expression, and thus improve the intestinal mucosal barrier function, especially in the PB group (Figure 2A,B). Similarly, ZO-1 mRNA expression in the colon tissue was significantly increased in the PC group, but not in the PB group, compared with the AB group (Figure 2C). In addition, compared with normal group, there was no significant difference in mRNA expression of occluding 1 (OC-1) in colon of AAD mice, while there was an increasing trend in the PC group (Figure 2D). The above results indicated that PCP can increase the expression of intestinal tight junction protein, especially ZO-1 protein, in AAD mice.

### 2.4. PCP Has the Potential to Maintain Intestinal Health for AAD Mice

#### 2.4.1. Effect of PCP on Microbial Operational Taxonomic Units (OTU) Number and Abundance

Dysregulation of intestinal flora is one of the main causes and pathological manifestations of AAD. In the current study, cecal contents were sequenced to analyze the changes of intestinal microbiota in mice with AAD, and the regulatory effect of PCP on intestinal microbiota was simultaneously studied. OTU are uniform symbols assigned to a taxon in phylogenetic or population genetic studies for the convenience of analysis, mainly including OTUs number and OTU components analysis. Venn diagrams can show the number of common and unique OTUs of multiple samples, and visualize the overlap of OTUs among samples. Venn diagram analysis of OTU level (Figure 3A) showed that 142 OTUs overlapped among the four groups at the cut-off condition of 97% sequence similarity, and 535, 284, 542 and 413 OTUs were identified in the NR, AB, PC and PB groups, respectively. OTU Rank curve can simultaneously explain the richness and uniformity of species contained in the sample. The experimental results (Figure 3B) showed that, compared with the NR group, the total number of OUT and diversity of bacteria in the AB group were significantly reduced. Similarly, the intervention of PCP and probiotics significantly increased the abundance and evenness of intestinal microbiota in AAD mice; especially the PC group had basically recovered to the same level as the NR group. To more visually demonstrate the dispersion and aggregation of the microbiota among the groups, OTU PCA (principal components analysis) and OTU PLS-DA (partial least squares discrimination analysis) were used to observe the similarity and difference of microbiota between groups, respectively. The results (Figure 3C,D) showed that the species distribution in AB and PB groups was far away from the NR group, while the PC and NR groups were basically clustered together. These findings suggest that PCP can improve the intestinal microbiota composition, increasing the abundance and diversity of microbiota in AAD mice.

#### 2.4.2. Effect of PCP on Microbial Diversity

Alpha diversity, describing the biodiversity within a particular region or ecosystem by analyzing the species diversity in individual groups, mainly includes six indexes, namely observed species, chao, ace, Shannon, Simpson and good-coverage indexes. The complexity of the sample is proportional to the observed species, chao, ace, and Shannon indexes, with a negative correlation with Simpson value. The good-coverage index is positively correlated with the integrity of sample detection, representing the credibility of the sequencing data. The results of alpha diversity (Figure 4A) showed that group AB had the lowest values in the observed species, chao, ace and Shannon, and the highest value in Simpson, which indicated that the AB group has the lowest diversity of species. In the indexes of observed species, chao, ace, and Shannon, the values from high to low were as follows: group NR > group PC > group PB > group AB, which suggested the species richness order of the four groups. The value of the Simpson index between groups was opposite to the above four indexes, which further confirmed the results of species diversity. The good-coverage indexes of all groups were greater than 0.999 (Figure 4A), indicating that the sample sequencing results had good integrity and reliability.

Beta diversity analysis was used to compare the differences in species diversity among samples, generally measured by several indexes, such as Bray-Curtis, weighted UniFrac, Unweighted UniFrac and Pearson. The beta diversity results of cluster tree and Heatmap (Figure 4B,C) found that NR group and PC group were similar, while that was significantly different from AB group and PB group, further demonstrating that the PCP has a certain influence on the community composition of microbiota.

#### 2.4.3. Effect of PCP on the Composition and Structure of Intestinal Flora

To obtain the species classification information corresponding to each OTU, the RDP classifier Bayesian algorithm was used to perform taxonomic analysis on the OTU representative sequences, and the community composition of each sample was counted at the level of kingdom, phylum, class, order, family, genus and species. The species bar chart can visually show the composition and proportion of species in each group, reflecting the changes of species among groups. Microbiota in gut include two predominant phylums related to metabolic diseases, *Bacteroidota* and *Bacillota*.

*Bacillota* and *Bacteroidota* are the dominant groups in the gut microbiota. *Bacillota,* which account for about 70% of the phylogenetic diversity of the gut microbiota, are the largest and most diverse bacterial group in the gut microbiota and are particularly important for the production of short-chain fatty acids (SCFAs, e.g., butyrate). Figure 5A,B shows the relative abundance of intestinal mucosal bacteria in mice at the phylum level, in which four taxa, namely, *Bacillota, Bacteroidota, Pseudomonadota* and *Verrucomicrobiota*, were the dominant phylum and accounted for a large proportion. Compared with the NR group, the relative abundance of *Verrucomicrobiota* and *Pseudomonadota* increased, while that of *Bacillota* decreased in the AB group, indicating that antibiotics significantly boost *Verrucomicrobiota* and *Pseudomonadota*, inhibit *Bacillota*. In contrast, compared with the AB group, the relative abundance of *Bacillota* increased, while that of *Verrucomicrobiota* and *Pseudomonadota* decreased after treatment with PCP and probiotics. The relative abundance of *Bacteroidota* changed slightly, but there was no significant difference between groups. Figure 5C,D shows the relative abundance of intestinal flora in mice at the genus level. *Allobaculum, Parabacteroides, Akkermansia, Desulfovibrio, Clostridium*, and *Lactobacillus* are the dominant genera with a large proportion. Compared with the NR group, the relative abundance of *Parabacteroides, Akkermansia, Desulfovibrio, Clostridium*, and *Sutterella* increased, while that of *Allobaculum, Ruminococcus*, and *Turicibacter* decreased in the AB group. Accordingly, the relative abundance of related bacterial genera in AAD mice returned to normal levels after treatment with PCP and probiotics.

To explore the characteristic microbiota species associated with diseases and drugs, we further analyzed the specific changes of gut microbiota at the species level in detail. According to the statistical analysis of the results (Figure 5E,F), it was found that PCP alleviated the symptoms of AAD mice by restoring seven characteristic species of intestinal tract, including the following species: *Parabacteroides_distasonis, Akkermansia_muciniphila, Clostridium_saccharolyticum, Ruminococcus_gnavus, Lactobacillus_salivarius, Salmonella_enterica and Mucispirillum_schaedleri*. These results suggest that PCP can effectively alleviate the imbalance of bacterial flora abundance in AAD mice, with a greater advantage than probiotics. 

Heatmaps of species abundance, with color gradients representing the relative abundance size of species and clustering according to species or sample abundance similarity, can visually show the similarity and difference of sample clustering and community composition among groups. Here, the results of heatmap and PCA analysis (Figure 6) showed that there was a significant difference between the abundances of bacterial species in group AB and group NR, indicating that antibiotics caused abnormal abundances of bacterial species in AAD mice.

#### 2.4.4. Predicted Functional Changes in Microbiota and Correlation Network Analysis

LEfSe analysis was performed to further verify the specific microbiota associated with PCP and visually demonstrate the degree of association among species (Figure 7A,B). To further understand how antibiotics modulate the changes of the gut microbiota in AAD mice, Phylogenetic Investigation of Communities by Reconstruction of Unobserved States (PICRUSt) analysis was used to predict microbial functions based on 16S rRNA sequencing. Accordingly, PICRUSt mapped 16S sequencing results with functional gene compositions in clusters of orthologous groups of proteins (COG) database to obtain functional prediction results, including COG_ level1 and COG_level2. At COG_level2, the enrichment results showed that, compared with the NR group, intestinal microbiota dysregulation in AAD mice could affect various biological functions, including amino acid transport and metabolism, carbohydrate transport and metabolism, nucleotide transport and metabolism, replication/recombination and repair functions, and defense mechanisms. These biological processes are mainly related to short-chain fatty acid metabolism and intestinal immune defense function, suggesting that antibiotics-induced intestinal microbiota disorder could lead to some secondary diseases.

Similarly, PICRUSt was used to obtain three levels of metabolic pathway information, based on Kyoto Encyclopedia of Genes and Genomes (KEGG) database information and OTU abundance information. At COG_level2 (Figure 7C), compared with the NR group, the relative abundance of some gut microbiota was significantly decreased in AAD mice, mainly involved in carbohydrate metabolism, amino acid metabolism, nucleotide metabolism, energy metabolism, immune system, replication and repair. In addition, the relative abundance of other microbiota was significantly increased, mainly involved in cardiovascular diseases, neurodegenerative diseases, the circulatory system, endocrine system and digestive system. The result (Figure 7D) indicates that, on the one hand, intestinal microecological imbalance and diarrhea may only be the initial symptoms of taking antibiotics, and on the other hand, the damage to the health of multiple systems could be the longer-term consequences. In conclusion, we hypothesized that the symptoms in AAD mice could blame on antibiotic-induced microbiota imbalance, as well as metabolic and immune dysfunction. Interestingly, some previous studies had demonstrated that PCP was beneficial to body health by regulating short-chain fatty acid metabolism and intestinal immunity through intestinal flora.

### 2.5. Effect of PCP on the mRNA Expression of Inflammation Indicators and SCFAs Receptor in Colon

Based on the previous results, it is believed that antibiotic-induced diarrhea is mainly related to the metabolism and immune function of intestinal flora. Therefore, the expression levels of nuclear factor kappa-B (NFKB), FOXP3, GPR41 and G protein-coupled receptor 43 (GPR43) in colon tissues were detected (Figure 8). Compared to mice in the NR group, mice in the AB group had a moderately reduced expression of FOXP3, GPR41, NFKB and GPR43, but with no significant differences for the latter two. Compared to mice in the AB group, mice in the PC group promoted the expression of the above four genes simultaneously, which restored the expression to the normal level or even higher. The results showed that PCP could activate the immune response and promote the expression of SCFAs receptor in colon tissue to a certain extent. However, the intervention of probiotic did not have a similar regulatory effect as PCP.

## 3. Discussion

In recent years, the extensive use of antibiotics has caused to the increased incidence of antibiotic-associated diarrhea [20]. Clinical [21,22] and animal [23] experiments have confirmed that antibiotic treatment significantly changes the structure of gut microbiota and reduces the diversity of gut microbiota. Commonly, antibiotic-associated diarrhea is closely associated with antibiotic-induced dysbiosis of the gut microbiota [9]. Consistent with the results of previous studies, the combination of antibiotics in this study can lead to intestinal microecological imbalance, significantly change the structural composition of intestinal microbiota, reduce the diversity of intestinal microbiota, and impair intestinal mucosal barrier function in AAD mice.

The current treatment of AAD mainly focuses on the elimination of pathogenic bacteria, and often ignores the root cause of intestinal microbiota disorder. The re-use of antibiotics in the treatment of AAD is likely to cause a new dysregulation of intestinal flora and even increase the incidence of critical illness [12]. According to traditional Chinese medicine, AAD mostly belongs to spleen deficiency and dampness, leading to digestive and absorption disorders, and the destruction of the balance between organs, resulting in the imbalance of bacteria, poor appetite, loose stool, emaciation and other symptoms. Therefore, the holistic concept of traditional Chinese medicine and the treatment concept of syndrome differentiation are more reasonable, which can reduce the adverse reactions caused by western medicine, and which has important clinical significance and good application prospects [12]. At present, traditional Chinese medicine, especially herbal polysaccharides with beneficial effects on Qi and spleen, is a hot topic in the treatment of AAD by regulating intestinal flora [6,17,19,24]. Our study revealed that PCP can comprehensively improve the clinical symptoms of AAD mice, including fecal traits, mental state, hair quality, etc., similar to the effect of probiotics. These effects may be attributed to the tonic function of PCP, while visually demonstrating its prebiotic effects.

Herbal polysaccharides regulate the balance between various flora to play a role in the treatment of diseases by promoting the growth of intestinal beneficial bacteria, inhibiting the growth of harmful bacteria regulation, and improving the intestinal mucosal barrier. Polysaccharides, also known as polyglycans, are widely found in animals, plants and microorganisms. Most traditional Chinese medicines are administered orally by decocting in water, and soluble polysaccharides often account for a large proportion in the decoction of traditional Chinese medicines. Most polysaccharides, however, cannot be directly digested and absorbed by the human body due to the lack of polysaccharides hydrolase [25]. Studies have found that intestinal flora is an important pathway in the interaction between polysaccharides and the human body [25,26]. On the one hand, it can degrade polysaccharides, promote the absorption and use of polysaccharides by the human body, and produce active metabolites with pharmacological effects. On the other hand, the proportion of intestinal flora can be adjusted to increase beneficial bacteria and reduce harmful bacteria, so as to improve the health of the body.

Researchers found that ganoderma lucidum polysaccharide could significantly increase the abundance of intestinal microorganisms, reduce the ratio of *Bacillota* to *Bacteroidota*, increase the level of serum cytokines, and strengthen the intestinal barrier in mice [27]. From intestinal photos and pathological observation, PCP significantly improved the substantial structure of the intestine of AAD mice, and the overall effect was better than that of probiotics. In addition, we further found that PCP improved the intestinal barrier defense function of AAD mice by increasing the expression levels of colonic tight junction protein ZO-1 and its mRNA. However, the expression of PCP and OC-1 was not found, indicating that PCP may have certain specificity in regulating intestinal tight junction proteins. Similarly, probiotics, similar to PCP, also promoted the expression of ZO-1 protein but had no effect on its mRNA expression level, indicating that the prebiotic effects of the two were different.

Intestinal microecology, as the largest microecosystem in the body, is mainly composed of *Bacillota*, *Bacteroidota*, *Pseudomonadota* and *Actinomycetota* among which *Bifidobacteria* and *Lactobacillus* are beneficial bacteria, while *Enterococcus* and *Fusobacteriota* are harmful bacteria. Changes in the proportion of intestinal flora are closely related to the occurrence and development of various diseases [5]. Different kinds of polysaccharides can increase good gut microbes and decrease bad ones. One study found that the ratio of *Bacillota* to *Bacteroidota* in the cecum of C57BL/6J mice increased significantly after 28 days of *Flammulina velutipes* administration [28]. Researchers studied the effects of *Ganoderma lucidum* polysaccharide and *Poria cocos* polysaccharide on the gut microbiota of C57BL/6J mice and found that two edible and medicinal fungal polysaccharides increased the production of SCFAs and lactate, as well as the abundance of anti-obesity probiotics, such as *Bifidobacterium*, *Eubacterium rectum*, *Lactobacillus* and *Lactococcus* [29]. Liu et al. found that an insoluble polysaccharide, extracted from the sclerotium of *Poria cocos*, can increase the abundance of *Trichoderma* and *Clostridium* butyrate in the cecum of obese mice, and then increase the content of butyrate in the intestine and improve the integrity of intestinal mucosa [13]. Meanwhile, the research team also found that oral administration of *Poria cocos* polysaccharide could reduce intestinal microbiota dysregulation and fungal overgrowth, significantly increase *Bacillota*/*Pseudomonadota* ratio, and increase the abundance of *Trichococcus*, thereby reducing liver inflammatory injury and fat accumulation in mice with alcoholic fatty liver disease (AHS) [30]. In our study, PCP not only increased the abundance of gut microbiota, but also increased the diversity of gut microbiota in AAD mice, including alpha diversity and beta diversity. Compared with the AB group, the relative abundance of *Bacillota* increased to the normal level, while that of *Verrucomicrobiota* and *Pseudomonadota* decreased after treatment with PCP and probiotics. Furthermore, we found that PCP may alleviate the symptoms of AAD mice by restoring seven characteristic species in gut, including the down-regulation of *Parabacteroides_distasonis, Akkermansia_muciniphila, Clostridium_saccharolyticum, Lactobacillus_salivarius, Salmonella_enterica, Mucispirillum_schaedleri*, and the up-regulation of *Ruminococcus_gnavus.* According to a new study, the colonization of *Parabacteroides_distasonis* may accelerate the occurrence and development of Type 1 diabetes in NOD mice by increasing macrophages, dendritic cells and damaging CD8+ T cells, and reducing Treg cells [31]. Another study also showed that mice with Crohn’s disease developed depression-like behavior during the chronic inflammatory phase, accompanied by an increase in the abundance of *Parabacteroides_distasonis* in the gut microbiota [32]. A clinical study, involving the microbiome of 71 untreated MS patients and 71 healthy controls, found increases in both *Akkermansia muciniphila* and *Acinetobacter calcoaccius* in multiple sclerosis patients, inducing proinflammatory responses in human peripheral blood mononuclear cells and monoclonal mice [33]. *Clostridium_saccharolyticum* has a well-known role in antibiotic-associated diarrhea, but in addition, its persistent infection may increase the risk of developing inflammatory bowel disease [34]. Studies have shown that *Salmonella enterica* invades the intestinal epithelium in the ileum and colon, causing an eosinophilic gastroenteritis, which usually causes fever, abdominal pain, vomiting and diarrhea, or enters the bloodstream and spreads throughout the body, causing sepsis [35]. A recent study, in an experiment with an unusual *Salmonella* infection, found that a low-abundance murine symbiotic strain, *Mucispirillum Schaedleri*, could help the host resist *Salmonella typhimurium* infection by competing for nutrients [36]. The decreased abundance of the above specific functional flora may be involved in the regulation of intestinal immunity and inflammation by PCP. Notably, *Ruminococcus_gnavus*, a commensal bacterium in the human gut, is commonly found in increased abundance in inflammatory bowel diseases and neurological diseases [37]. The colonization of *Ruminococcus_gnavus* significantly reduced PSA-NCAM+ granulocytes and increased microglia recruitment in the dentate gyrus of the hippocampus in mice [37]. In Pan’s cell-specific lysozyme knockout (Lyz1) mice, transplanting *Ruminococcus_gnavus* can induce type 2 immune response, skin reprogramming, and enhance the anti-colitis ability [38]. Another study has suggested that *Ruminococcus_gnavus* may express two “superantigen” proteins, including IbpA and IbpB, which can bind to the variable region of IgA and activate the IgA response in mice, playing an important role in intestinal homeostasis [39]. *Ruminococcus* was inversely correlated with IgA antibody titers of rotavirus and norovirus, revealing the potential as antiviral bacteria. Similarly, the abundance of *Ruminococcus_gnavus* has increased significantly in our study and may play an important regulatory function [40]. Finally, through data enrichment analysis, we found that PCP may affect the intestinal mucosal barrier function, host immune response and metabolic function by regulating the microbiota, so as to achieve the therapeutic effect of AAD.

PCP has been shown to have important biological regulatory activities, including t anti-inflammatory activity, anti-inflammatory activity, and modulation on gut microbiota. It was discovered by Lu et al., notably, that polysaccharides, isolated from cultured mycelia of *Poria cocos*, has a dose-dependent effect in inhibiting the production of the interferon (IFN)-γ-induced inflammation marker IP-10, which suggests the PCP-II to be a promising anti-inflammatory agent [41]. The anti-inflammatory activity and underlying mechanisms of PCP in RAW 264.7 cells were investigated, suggesting PCP exerts anti-inflammatory effects by inhibiting LPS-stimulated overproduction of NO, IL-6, TNF-α and IL-1β [42]. Meanwhile, PCP has been reported to inhibit the TLR4/NF-κB pathway, thereby preventing the development of AS [43]. By contrast, compared with the normal group, the levels of nitric oxide, IL-2, IL-6, IL-17A, TNF and IFN-γ were increased in the PCP-treated mice, indicating that PCP can activate the TLR4/TRAF6/NF-κB signaling pathway to mediate immunomodulatory activity [44]. These inconsistent results suggest that the immunomodulatory activity of PCP is bidirectional and dynamic, and has a corresponding specific regulatory trend in different pathological stages of the disease. In future experiments, focusing on the intervention of PCP in different periods may help to elucidate the mechanism of bidirectional immune regulation. Our preliminary research shows that PCP could activate the immune response by promoting the mRNA expression of FOXP3 and NFKB, even higher than in the normal group. The results suggest that PCP may have the function of activating immune response in both healthy and diseased states. In addition, based on the regulatory effect of PCP on promoting the expression level of SCFAs receptor in colon tissue, we speculated that the changes of microbial metabolites, especially SCFAs, may be the underlying mechanism of PCP prebiotics.

This study still has some limitations. Firstly, intestinal microecology includes intestinal structure, microorganisms and microbial metabolites, but this study has not explored the differential changes of microbial biota metabolites. Later studies should use metabolomics analysis to further explore the specific mechanism of PCP improving AAD symptoms by screening different metabolites. Secondly, this study screened out some different microbiota, which still need to be verified by microbiota transplantation and other means, and also looked for the direct dose-effect relationship between microbiota and metabolites. Moreover, the exploration of intestinal mucosal immunity in this study is still in the preliminary stage and needs to be further studied.

## 4. Materials and Methods

### 4.1. Animals and Grouping

Specific-pathogen-free C57BL/6N mice (*n* = 24, aged 5 weeks, weighing 20 g ± 2 g, half males and half females) were purchased from Guangdong Medical Laboratory Animal Center (Foshan, China). The experiment was performed under the supervision and evaluation of the Experimental Animal Ethics Committee of Jinan University (Guangzhou, China). All experimental procedures were strictly conducted in accordance with the Guidelines for the Care and Use of Laboratory Animals (ninth edition; National Institutes of Health, Bethesda, MD, USA) and were approved by the Animal Ethics Committee of Jinan University (Approval No.20190312).

All mice were maintained in a 12 h light/dark cycle and conventional housing conditions (relative temperature 22 °C ± 1 °C; relative humidity 50% ± 5%), and had free access to food and water. Mice were randomly divided into four groups (*n* = 6):the normal group (NR), antibiotic-associated diarrhea group (AB), *Poria cocos* polysaccharides treatment group (PC) and probiotics treatment group (PB). 

### 4.2. Drugs and Modeling

*Poria cocos* polysaccharides (product code: S25585) was bought from Shanghai Yuanye Bio-Technology Co., Ltd. (Shanghai, China). Probiotics were provided by Bifico capsules (approval number: S10950032) purchased from Shanghai Sinepharm (Shanghai, China), which combines Bifidobacterium longum (>1.0 × 10^7^ CFU), *Lactobacillus acidophilus* (>1.0 × 10^7^ CFU) and *Enterococcus faecalis* (>1.0 × 10^7^ CFU) in each capsule (210 mg).

For the induction of antibiotic-associated diarrhea, all mice, except those in the normal group, were treated with broad-spectrum antibiotics in drinking water for seven days (ampicillin, 1 g/L, Sigma-Aldrich (Shanghai, China); neomycin sulfate, 1 g/L, Sigma-Aldrich (Shanghai, China); metronidazole, 1 g/L, Sigma-Aldrich (Shanghai, China); vancomycin, 500 mg/L, Sigma-Aldrich (Shanghai, China)) [6,23] and treated as shown in Figure 9. Then, the groups were given their respective drugs orally administered daily for seven consecutive days using intragastric gavage. The mice in the probiotics treatment group (PB) were given Bifico capsules (4.2 g/kg/d) [45] while mice in the *Poria cocos* polysaccharides treatment group (PC) were given PCP (250 mg/kg/d) [46].

### 4.3. General Condition Observation

The general conditions, including body weight, food/water intake, activity, mental state, stool status, survival, anal prolapse, and other complications, were observed and compared before and after modeling and after administration. The general condition was observed daily, and the body weights were measured every two days.

### 4.4. Sample Collection and Storage

All mice were anesthetized with isoflurane and sacrificed by exsanguination. The intestines were removed and pictures were taken for a visual record. Part of the colon and small intestine was fixed in 4% paraformaldehyde for examination of histopathology, and the rest was collected for subsequent experiments after cleaning the contents with PBS. Cecum content was collected in sterile EP tubes. All samples were stored at −80 °C until use.

### 4.5. Examination of Histopathology

The colon and small intestine fixed in 4% paraformaldehyde were embedded, sliced into 5 μm thickness, and then hematoxylin-eosin staining and immunofluorescence staining were performed according to the protocol (Appendix A). Histopathological changes were observed by the ortho-fluorescent and microscopy imaging system (Nikon, Tokyo, Japan), and then quantified using Image-Pro Plus software (Media Cybernetics, Rockville, MD, USA).

### 4.6. Gut microbiota Analysis

#### 4.6.1. Genomics DNA Extraction for 16S rRNA Sequencing

The microbial community DNA was extracted for the stool samples collected from the cecum using the QIAamp Fast DNA Stool Mini Kit (Cat: 51604, Qiagen, Hilden, Germany), following the manufacturer’s instructions. DNA was quantified with a Qubit Fluorometer using a Qubit dsDNA BR Assay kit (Invitrogen, Carlsbad, CA, USA) and the quality was checked by running aliquot on 1% agarose gel.

#### 4.6.2. Sequencing and Bioinformatics Analysis

The PCR reaction system was configured with 30 ng of qualified genomic DNA sample and corresponding fusion primers, and PCR reaction parameters were set for PCR expansion. The PCR amplification products were purified and dissolved in an elution buffer using Agencourt AMPure XP magnetic beads (Beckman Coulter, Indianapolis, IN, USA), and labeled to complete the library construction. The Agilent 2100 Bioanalyzer (Agilent Technologies, Waldbronn, Germany) was used to detect the fragment range and concentration of the library while the HiSeq platform was selected for sequencing according to the size of the inserted fragments. Clean data with high quality were screened out from the database data for subsequent analysis. Reads are spliced into tags through overlap relationship between reads. The tags were clustered into OTU and compared with the database, and then species annotation was carried out. Based on OTU and annotation results, species complexity analysis of samples, species’ differences between groups, association analysis and model prediction were performed. The procedure and details of the operation are described in the Appendix A. 16S rRNA gene sequencing was carried out by Shenzhen BGI Gene Research Institute (Shenzhen, China).

### 4.7. Real-Time Reverse Transcription-Polymerase Chain Reaction (RT-PCR)

Total RNA was extracted by RNAiso Plus (TaKaRa, Maebashi, Japan) according to the manufacturer’s instructions. The cDNA synthesis and RT-qPCRs were conducted following related methods described in our previous article [47,48]. The primers were synthesized by Generay Biotech (Shanghai, China), and the primers for RT-qPCR are listed in Table 1.

### 4.8. Statistical Analysis

Statistical analyses were carried out using SPSS Statistics 25.0 (IBM Software, New York, NY, USA) and graphs were drawn using GraphPad Prism 9 (GraphPad Software, San Diego, CA, USA). All measurement data were presented as Mean ± SEM. Comparisons between groups were performed using unpaired *t*-tests, while comparisons involving multiple groups were analyzed using a one-way ANOVA. A *p*-value of <0.05 was considered to be statistically significant.

## 5. Conclusions

These results indicated that *Poria cocos* polysaccharide may ameliorate antibiotic-associated diarrhea in mice by regulating the homeostasis of the gut microbiota and intestinal mucosal barrier. In addition, polysaccharide-derived changes in intestinal microbiota were involved in the immunomodulatory activities and modulation of the metabolism.

## Figures and Tables

**Figure 1 ijms-24-01423-f001:**
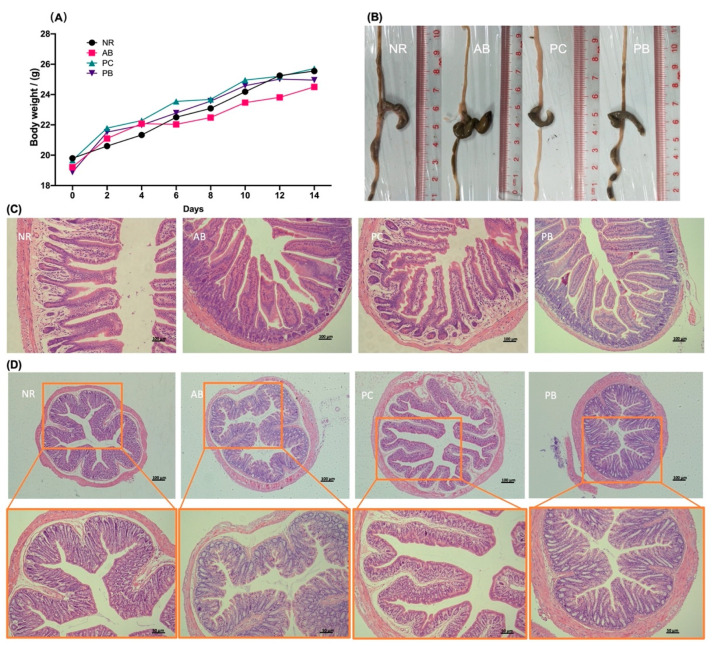
Observation of animal body weight and intestine. (**A**) Changes in body weights. (**B**) Macroscopic observation of intestine and stool status. (**C**) PCP treatment alleviates small intestine damage (scale bar = 100 µm). (**D**) PCP treatment alleviates colon damage (scale bar above = 100 µm and scale bar below = 50 µm). NR: normal group, AB: antibiotic-associated diarrhea group, PC: *Poria cocos* polysaccharides treatment group, PB: probiotics treatment group.

**Figure 2 ijms-24-01423-f002:**
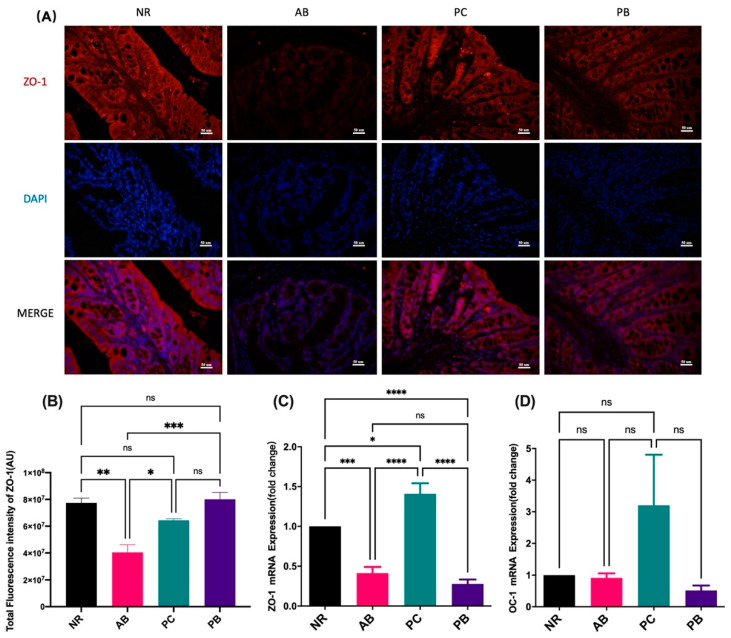
Expression of tight junction proteins ZO-1 and OC-1 in colon tissue (*n* = 5). (**A**,**B**): immunofluorescent staining and analysis of the tight junction protein ZO-1 in colon (scale bar = 50 µm). (**C**,**D**): mRNA expression of ZO-1 and OC-1 in colon. Data are shown with mean ± SEM, * *p* < 0.05, ** *p* < 0.01, *** *p* < 0.001, **** *p* < 0.0001, ns = no significance.

**Figure 3 ijms-24-01423-f003:**
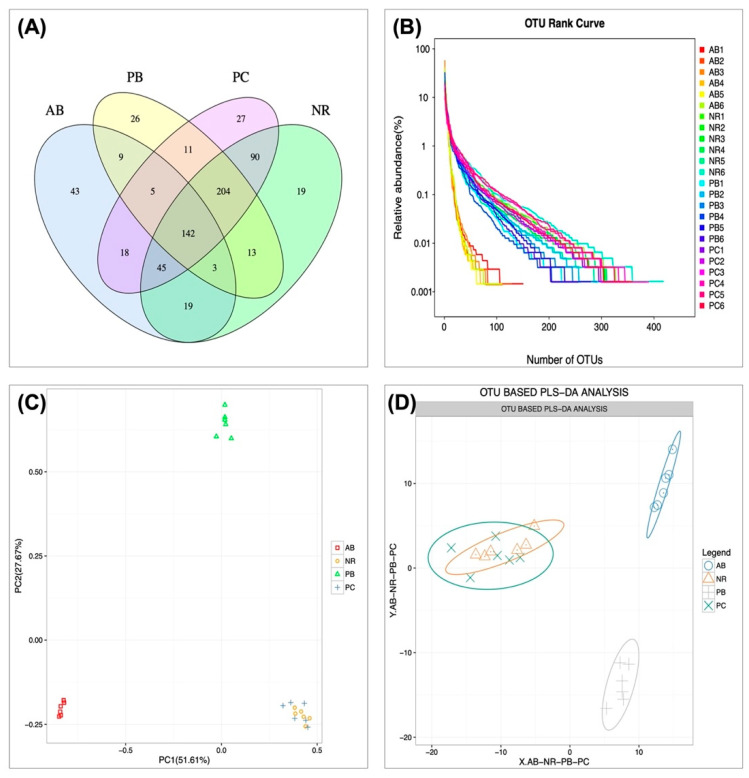
OTU cluster and analysis (*n* = 6). (**A**) Venn diagram: the distribution of the number of intestinal mucosal bacterial OTUs in each group of samples. (**B**) OTU rank curve of each sample. (**C**) PCA based on OTU abundance. (**D**) PLS-DA based on OTU abundance.

**Figure 4 ijms-24-01423-f004:**
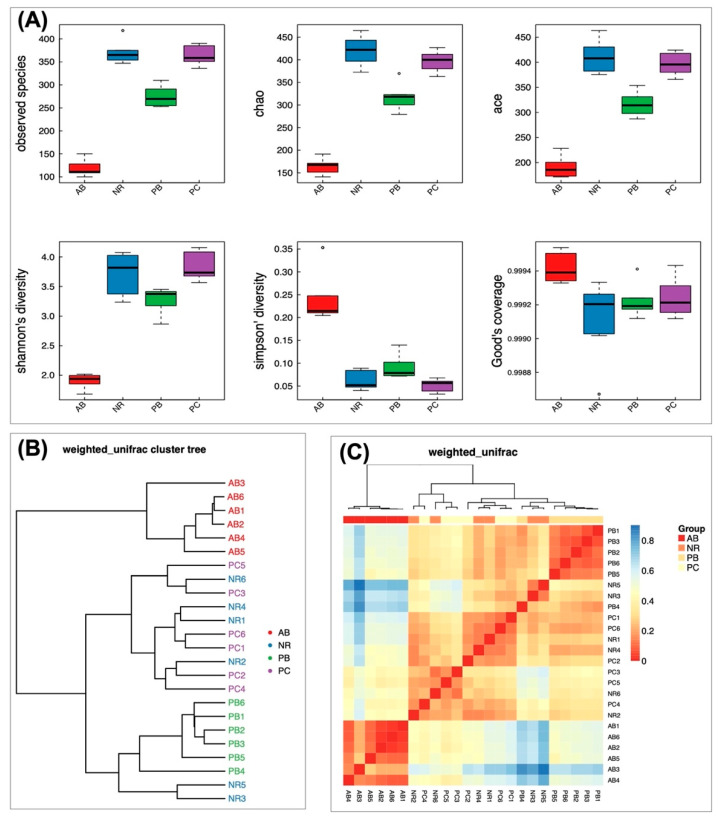
Diversity analysis. (**A**) Boxplots of the observed species, chao, ace, Shannon, Simpson and good-coverage indexes. The five lines from bottom to top are: minimum, first quartile, median, third quartile and maximum. Outliers are denoted by dots “o”. (**B**) Sample clustering tree. Samples with the same color in the figure belong to the same group. The closer the samples were, the more similar the species compositions of the two samples were. (**C**) Heatmap of beta diversity. The larger the index, the greater the difference between samples. The closer the trees are, the more similar the samples are.

**Figure 5 ijms-24-01423-f005:**
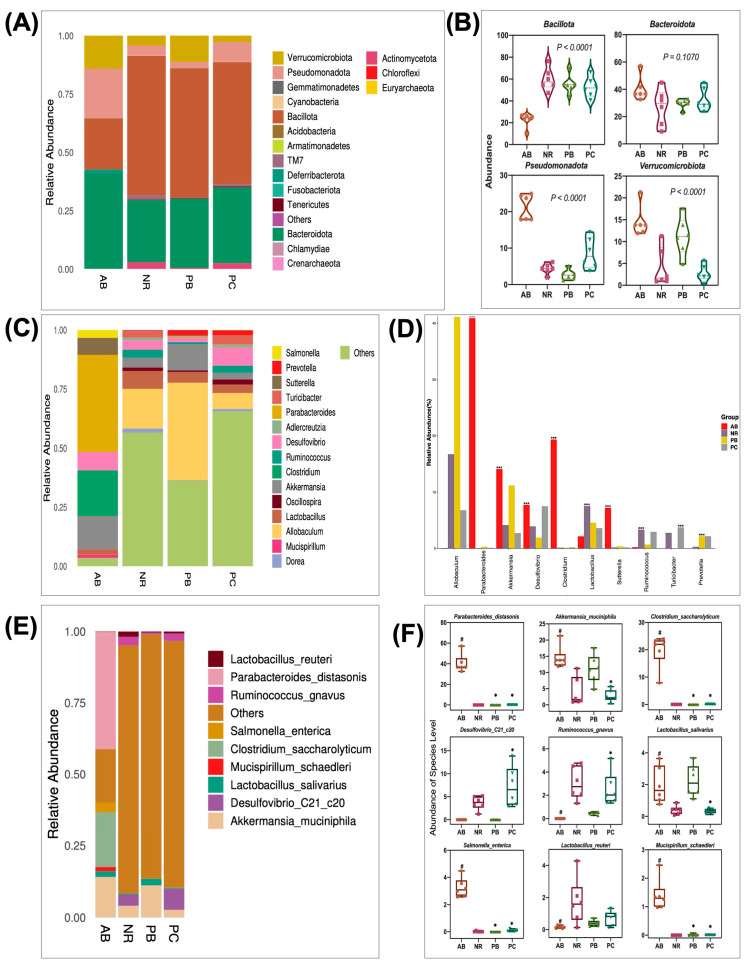
Species composition and abundance of intestinal flora (*n* = 6). The taxonomic composition distribution in samples at the phylum level (**A**), genus level (**C**), and species level (**E**). Statistical results of microflora with significant changes at the phylum level (**B**), genus level (**D**), and species level (**F**). Data are shown with mean ± SEM, ^#^
*p* < 0.05 (compared to NR group), * *p* < 0.05 (compared to AB group), *** *p* < 0.001.

**Figure 6 ijms-24-01423-f006:**
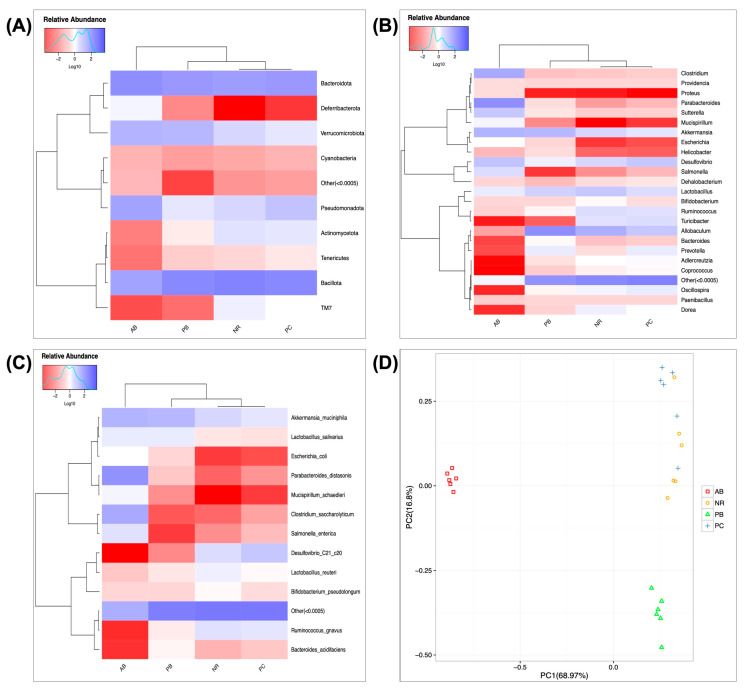
Heatmaps of species abundance at the phylum level (**A**), genus level (**B**), and species level (**C**), and the PCA analysis of species abundance (**D**) (*n* = 6). The relative abundance of species was log transformed to eliminate the effect of too-large-difference in the relative abundance of species on sample clustering. In the PCA analysis, the distance between samples was negatively correlated with the degree of similarity in species composition.

**Figure 7 ijms-24-01423-f007:**
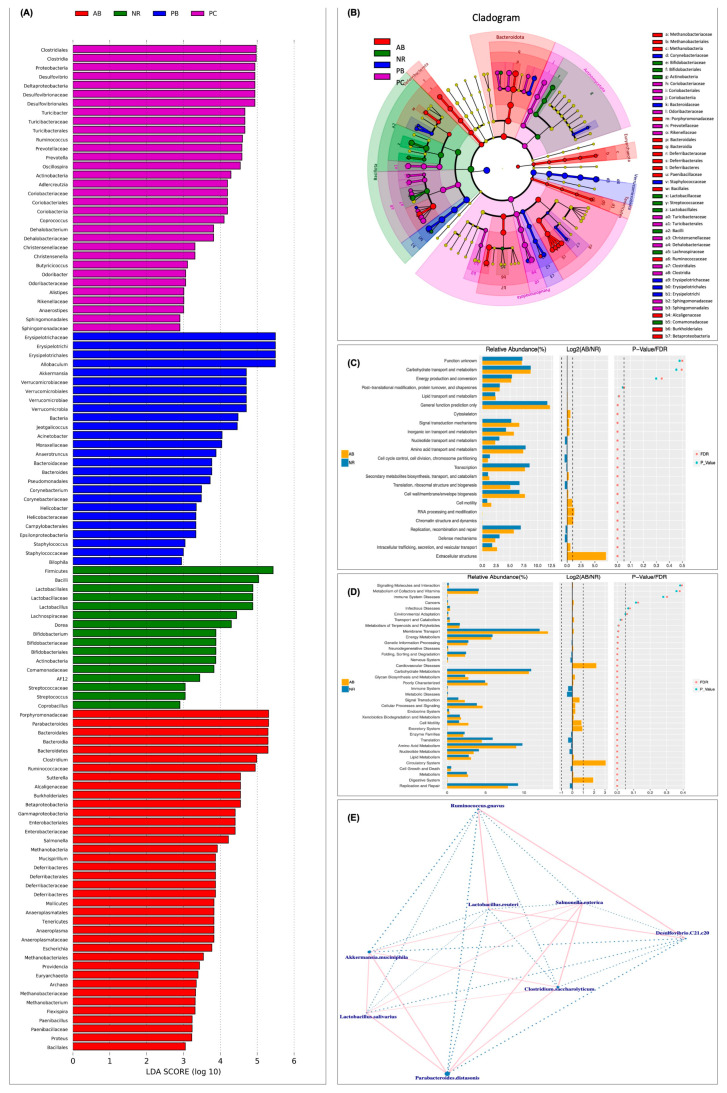
Biomarker analysis and functional prediction. (**A**) LDA (linear discriminant analysis) score plot based on LEfSe analysis. It mainly shows significantly different species (biomarker) with LDA score greater than the preset value. Different colors represent significant microbial groups in different groups. (**B**) Cladogram of LEfSe analysis. Colors represent groups while the color circle indicates a biomarker. From the inside to the outside, each circle in turn is phylum, class, order, family, genus level species. (**C**) Functional analysis of biomarkers of COG_level2. (**D**) KEGG pathways enrichment analysis of level 2. From left to right, the figure shows the relative abundance of pathways, the log2 value of the mean ratio of the relative abundance of two groups of the same pathway, and the *p*-value and false discovery rate (FDR) value obtained by Wilcox test. *p*-value and FDR values less than 0.05 indicate significant differences in this pathway between the two groups. (**E**) Correlation network analysis of microbial species level. Species are connected using straight lines, pink for positive correlations, blue for negative correlations, and the thickness of the lines for the magnitude of the correlation.

**Figure 8 ijms-24-01423-f008:**
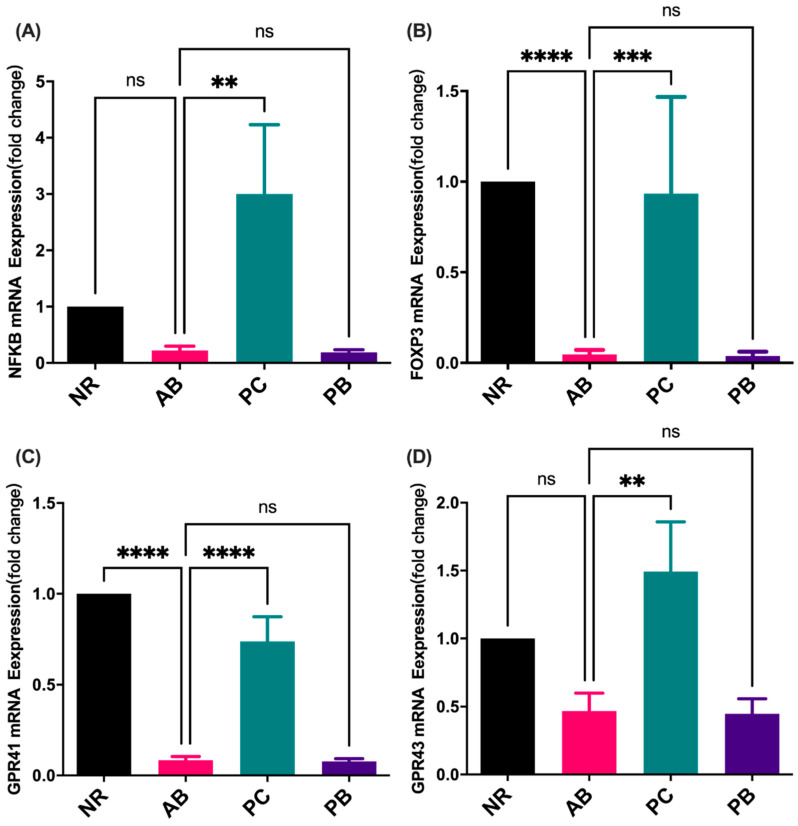
Colon gene expression (*n* = 5). Relative mRNA levels of the NFKB (**A**), FOXP3 (**B**), GPR41 (**C**), and GPR43 (**D**). Data are shown with mean ± SEM, ** *p* < 0.01, *** *p* < 0.001, **** *p* < 0.0001, ns = no significance.

**Figure 9 ijms-24-01423-f009:**
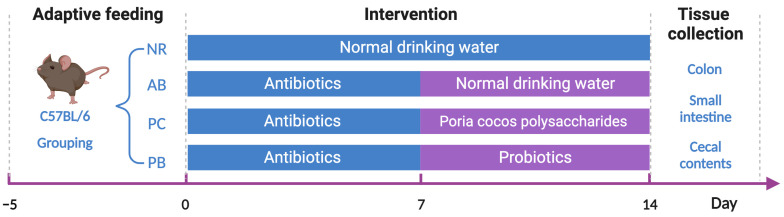
Flow chart for experiments with associated timing.

**Table 1 ijms-24-01423-t001:** The primers for RT-qPCR.

Gene Name	Primers (5′-3′)
ZO-1	Forward: GAGTGGACTATCAAGTGAGCCTAA Reverse: ATCCAAGTTGCTCGTCAATCTAA
OC-1	Forward: CTATGGGACAGGGCTCTTTGGA Reverse: AGGAAGCGATGAAGCAGAAGGC
NFKB	Forward: ATTCTGACCTTGCCTATCTAC Reverse: TCCAGTCTCCGAGTGAAG
FOXP3	Forward: CTCTAGCAGTCCACTTCACCAA Reverse: CACCCACCCTCAATACCTCTCT
GPR41	Forward: TCCTGCGGTCCACTCTTT Reverse: TTCCTCCAAGTTCCAAGC
GPR43	Forward: GCAGGGAGCCCAGTAAGA Reverse: TGCCCAAGGAGAATGACC
GAPDH	Forward: TGATGACATCAAGAAGGTGGTGAAG Reverse: TCCTTGGAGGCCATGTAGGCCAT

## Data Availability

The data used to support the findings are available from the corresponding author upon request.

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
