# Peer review of "Poria cocos Polysaccharide Ameliorated Antibiotic-Associated Diarrhea in Mice via Regulating the Homeostasis of the Gut Microbiota and Intestinal Mucosal Barrier"

_ijms, 2023, doi:10.3390/ijms24021423_

Round 1

Reviewer 1 Report

Line 12: Poria cocos italic.

Line 37: Poria cocos italic.

Line 103: Poria Cocos italic and cocos small letter.

 Line 112-115: 7  replace seven.

Line 117: Poria Cocos, cocos small letter.

 Line 481: poria cocos, Poria capital letter. 

Line 81: Poria Cocos, cocos small letter. 

Add a list of abbreviations: word lines, 63,68,78,103,104,238,358,367,368,384,389, 392,398,402.

Supplemental Methods and Materials: write numbers 1-10 in letters. 

Author Response

Dear reviewer, thank you for your valuable comments.

Cover letter and Replies to Editor and Reviewers

2022.12.28

Thank you for your letter and for the reviewers’ comments concerning our paper entitled: “Poria cocos polysaccharide ameliorated Antibiotic-Associated Diarrhea in Mice via regulating the homeostasis of the gut microbiota and intestinal mucosal barrier” (ID: ijms-2061360). We have studied the comments carefully and have made correction which we hope meet with approval. Revised portion are marked up using the “Track Changes” function in MS Word.

The main corrections in the paper and the responds to Editor and the reviewer’s comments are as flowing:

Replies to Review Report (Editor)

  1. Structure of "Abstract" is incorrect; please delete "Background", "Methods", "Results", "Conclusions" from abstract.

Dear editor, we have deleted "Background", "Methods", "Results", "Conclusions" from abstract as suggested.

  1. Please change the name of all citated phylum to the current one according to the NCBI taxonomy (i.e., Firmicutes is now Bacillota etc.)

Dear editor, thank you for your valuable advice! That provided us with valuable learning opportunities and prompted us to update our understanding of the current names of intestinal flora.

According to your suggestion, we have changed the name of all citated phylum, including that in the manuscript and figures, to the current one according to the NCBI taxonomy. Revised portion are marked up using the “Track Changes” function in MS Word. Thank you again.

Replies to Review Report (Reviewer #1)

  1. Poria cocos” should be written in italic Poria cocos” and the numbers 1-10 should be written in letters in the manuscript and Supplemental Materials. 

Dear reviewer, thank you for your rigorous review and scientific spirit of pursuing perfection. We apologize earnestly for these mistakes in spelling in the manuscript, and we have made corrections one by one according to your comments.

  1. Add a list of abbreviations: word lines, 63,68,78,103,104,238,358,367,368,384,389, 392,398,402.

Dear reviewer, we apologize earnestly for the confusion caused by unclear abbreviations to you and readers. Based on your valuable suggestions, we have read through the full text and added a list of abbreviations at the end of the article as requested. Thank you for your valuable comments.

Replies to Review Report (Reviewer #2)

  1. These authors indicated that Poria cocos polysaccharide may ameliorated Antibiotic-Associated Diarrhea in Mice via regulating the homeostasis of the gut microbiota and intestinal mucosal barrier. which involved in the immunomodulatory activities and modulation on metabolic. The conclusion of this article is still meaningful,but there are still problems that need to be modified and improved. For example, the resolution of all the pictures in the article is not very clear, and many of them cannot be seen clearly. It is recommended to provide high-definition photos.

Dear reviewer, thank you for your valuable comments. We have provided high-resolution pictures on the word manuscripts submitted to the system, but the resolution of the pictures may be reduced because they are converted to PDF format. We are very sorry for the inconvenience caused to your review. Next, we will carefully communicate with the editor of the journal, and provide qualified pictures according to your and the editor's requirements.

  1. The scale of HE staining in the article is too small to see clearly, and the scale of immunofluorescence and other pictures is lacking, which needs to be added.

Dear reviewer, thank you for pointing out the ambiguity and absence of our scales in terms of pathological sections. We have made modifications, adjustments and supplements according to your suggestions. This is a careful but critical piece of advice. Thank you very much.

  1. In Fig.3, immunofluorescent staining and analysis of the tight junction protein Occludin-1 in colon, which needs to be added.

Dear reviewer, as you mentioned, OC-1 and ZO-1 are both important intestinal tight junction proteins, which play a significant role in intestinal mucosal barrier. In this experiment, we first detected the mRNA contents of ZO-1 and OC-1 in colon by RT-PCR, and then did immunofluorescence staining according to the statistical results of mRNA. Since the mRNA expression of OC-1 was not significantly different, immunofluorescence staining was not performed. The expression of ZO-1 was significantly different, which was further verified by immunofluorescence staining, thus proving the change of intestinal mucosal barrier structure.

  1. In Fig.9, Only detect the Effect of PCP on the mRNA expression of inflammation indicators and SCFAs receptor in colon were not enough, Changes in these protein levels also need to be detected using Western Blotting.

Dear reviewer, based on the predicted results of intestinal flora sequencing, we preliminatively investigated the mRNA expression trend of colon inflammation indicators and short-chain fatty acid receptors by RT-PCR. On the one hand, this study focused on the effect of PCP on intestinal flora and intestinal structure, so Western Blotting has not been carried out to further explore its underlying mechanism. We believe that this will be an important plan for our next experiments to confirm the deep mechanism by which PCP affects the body through the bacterial community. On the other hand, there is evidence in existing studies that the relationship between gut flora and these inflammatory markers [1-3] and short-chain fatty acid receptors [4,5] is relatively clear. In addition, many studies have shown that the extract and polysaccharide of poria cocos can affect immunity and intestinal short-chain fatty acid metabolism through intestinal flora [6-8].

We are very grateful for the reviewer's valuable advice. In future experiments, we will study the specific biological mechanism in detail through Western Blotting, bacterial transplantation, sterile animals and other technologies.

Reference:

[1] Zhang B, Zhao C, Zhang X, et al. An Elemental Diet Enriched in Amino Acids Alters the Gut Microbial Community and Prevents Colonic Mucus Degradation in Mice with Colitis. mSystems. 2022;7(6):e0088322.

[2] Martin-Gallausiaux C, Garcia-Weber D, Lashermes A, et al. Akkermansia muciniphila upregulates genes involved in maintaining the intestinal barrier function via ADP-heptose-dependent activation of the ALPK1/TIFA pathway. Gut Microbes. 2022;14(1):2110639.

[3] van der Veeken J, Campbell C, Pritykin Y, et al. Genetic tracing reveals transcription factor Foxp3-dependent and Foxp3-independent functionality of peripherally induced Treg cells. Immunity. 2022;55(7):1173-1184.e7.

[4] de Vos WM, Tilg H, Van Hul M, Cani PD. Gut microbiome and health: mechanistic insights. Gut. 2022;71(5):1020-1032.

[5] Chun E, Lavoie S, Fonseca-Pereira D, et al. Metabolite-Sensing Receptor Ffar2 Regulates Colonic Group 3 Innate Lymphoid Cells and Gut Immunity. Immunity. 2019;51(5):871-884.e6. 

[6] Imran Khan, Guoxin Huang, Xiaoang Li, Waikit Leong, Wenrui Xia, W.L. Wendy Hsiao, Mushroom polysaccharides from Ganoderma lucidum and Poria cocos reveal prebiotic functions,

Journal of Functional Foods, 2018(41),191-201.

[7] Sun S, Wang K, Sun L, et al. Therapeutic manipulation of gut microbiota by polysaccharides of Wolfiporia cocos reveals the contribution of the gut fungi-induced PGE2 to alcoholic hepatic steatosis. Gut Microbes. 2020;12(1):1830693.

[8] Liu J, Liu L, Zhang G, Peng X. Poria cocos polysaccharides attenuate chronic nonbacterial prostatitis by targeting the gut microbiota: Comparative study of Poria cocos polysaccharides and finasteride in treating chronic prostatitis. Int J Biol Macromol. 2021;189:346-355.

We tried our best to improve the manuscript and made some changes in the manuscript. These changes will not influence the content and framework of the paper. We appreciate for Editors’ and Reviewers’ warm work earnestly, and hope that the correction will meet with approval.

Once again, thank you very much for your comments and suggestions.

Best Regards.

Yours Sincerely,

Li Deng

Reviewer 2 Report

1.  These authors indicated that Poria cocos polysaccharide may ameliorated Anti- biotic-Associated Diarrhea in Mice via regulating the homeostasis of the gut microbiota and intestinal mucosal barrier. which involved in the immunomodulatory activities and modulation on metabolic.  The conclusion of this article is still meaningful,but there are still problems that need to be modified and  improved.  For example, the resolution of all the pictures in the article is not very clear, and many of them cannot be seen clearly. It is recommended to provide high-definition photos.

2.  The scale of HE staining in the article is too small to see clearly, and the scale of immunofluorescence and other pictures is lacking, which needs to be added.

3. In Fig.3, immunofluorescent staining and analysis of the tight junction protein Occludin-1 in colon, which needs to be added.

4.  In Fig.9,   Only detect the Effect of PCP on the mRNA expression of inflammation indicators and SCFAs receptor in colon were not enough,  Changes in these protein levels also need to be detected using Western Blotting.

Author Response

Dear reviewer, thank you for your valuable comments. The point-by-point  response, Please see the attachment.

Round 2

Reviewer 2 Report

None

Author Response

Dear reviewer, thank you very much for your comments and suggestions. We are honored to have your patient guidance on our paper.